# Porphyrin Modified UiO-66-NH₂ for Highly Efficient Photoreduction of Cr(VI) under Visible Light

**Kaiwen Yuan [1], Bo Gong [1], Chundong Peng [1], Yanmei Feng [1], Yingmo Hu [1], Kai Chen [2], Daimei Chen [1,\*] and Derek Hao [3]**

[1] Engineering Research Center of Ministry of Education for Geological Carbon Storage and Low Carbon Utilization of Resources, China University of Geosciences, Xueyuan Road, Haidian District, Beijing 100083, China; 2103210004@email.cugb.edu.cn (K.Y.)

[2] Jiangsu Key Laboratory of Atmospheric Environment Monitoring, Collaborative Innovation Center of Atmospheric Environment and Equipment Technology, Pollution Control School of Environmental Science and Engineering, Nanjing University of Information Science and Technology, Nanjing 210044, China

[3] School of Science, Science, Technology, Engineering and Mathematics (STEM) College, Royal Melbourne Institute of Technology University, Melbourne, VIC 3000, Australia

\* Correspondence: chendaimei@cugb.edu.cn

**Abstract:** Cr(VI) is a common heavy metal pollutants present in the aquatic environment, which possess toxic and carcinogenic properties. In this study, a solvothermal reaction was used to prepare porphyrin (TCPP)-modified UiO-66-NH₂ (UNT). The UNT integrated adsorption and photocatalytics in the application for dealing with Cr(VI). The photocatalytic reduction activities of UNT for Cr(VI) were investigated under visible light illumination. We found that the TCPP doping amount of 15 mg UNT (15-UNT) had a 10 times higher reduction rate of Cr(VI) than pristine UiO-66-NH₂. The optimal 15-UNT photocatalyst demonstrated the highest photocatalytic activity, and Cr(VI) was completely removed within 80 min. In addition, the introduction of porphyrin not only enhanced the absorption of light but also enabled the transport of photogenerated electrons from porphyrin to UiO-66-NH₂, which promoted the separation of charge carriers. Furthermore, the effects of factors such as porphyrin content, pH and light source on the photocatalytic reduction performances of UNT were also explored. Overall, this work presented a possible relationship between the crystal structures and the performance of UNT.

**Keywords:** photocatalytic reduction; porphyrin; UiO-66-NH₂; hexavalent chromium





## 1. Introduction

With the development of human society, various heavy metal ions in industrial wastewater have caused great harm to both the environment and human health [1,2]. Among the many toxic metal ions, Cr(VI) is one of the common heavy metal pollutants in the environment, which mainly originates from the metallurgical, textile and pharmaceutical industries. Cr(VI) is a toxic and carcinogenic metal ion that causes the human body complications, such as liver function damage and lung congestion. The best method of eliminating hazardous Cr(VI) is to reduce it to less hazardous Cr(III) and precipitate it [3–5]. Nowadays, some techniques, such as adsorption, electrocatalysis and photocatalysis, have been used to eliminate Cr(VI) from wastewater [6–8]. Photocatalysis is considered to be one of the most efficient methods due to its many advantages, such as convenient operation, non-pollution and recycling. Thus, it has been widely employed in the environmental field.

Nowadays, several catalytic systems have been reported for the photoreduction of Cr(VI). The metal–organic framework (MOF) is a type of periodic porous crystal material formed by organic ligands and metal ions [9]. In particular, MOFs have received widespread attention due to their unique advantages, such as large specific surface area, high crystallinity, tunable framework structure and thermal stability [10]. Using MOFs

as photocatalysts not only preserves the nanostructure of the metal–oxygen clusters, but its abundance of metal nodes and variety of organic ligands also allow for fine-tuning and design at the molecular level. For instance, Wang et al. prepared MIL-125 (Ti) and $NH_2$-functionalized MIL-125 (Ti) via a simple solvothermal method and investigated their performance for the photocatalytic reduction of Cr(VI) [11]. Additionally, Liang et al. have reported an amino-modified MIL-68(In) (MIL-68(In)-$NH_2$), a targeted photoactive catalyst, which can effectively shift the light absorption edge to the visible region by briefly introducing the amino group into the organic ligand [12]. Consequently, MOFs have been widely used in the field of catalysis [13–15].

UiO-66 consists of $Zr^{4+}$ and dicarboxylic acid ligands, and has excellent stability over a wide pH range [16,17]. It has been widely reported as a very popular photocatalyst. Due to limited charge separation and low light utilization, the UiO-66 exhibits poor photocatalytic performance. In order to improve its photoactivity, two main processes are carried out to modify UiO-66. One involves using other semiconductors (such as g-$C_3N_4$ [18], BiOBr [19] and $ZnIn_2S_4$ [20]) to compound with UiO-66, forming a heterojunction structure. For example, Shen et al. synthesized RGO-UiO-66 by the self-assembly of electrostatically derived UiO-66-$NH_2$ with graphene followed by hydrothermal reduction [21]. Compared to UiO-66-$NH_2$, this nanocomposite showed enhanced photocatalytic activity in the reduction process of Cr(VI). Moreover, Yi et al. simply prepared Z-scheme g-$C_3N_4$/UiO-66 heterojunctions constructed by 3D UiO-66 and 2D g-$C_3N_4$ sheets via ball milling [6]. Cr(VI) reduction was also achieved with simulated wastewater prepared from tap water and lake water, simulating seawater and leather tannery wastewater.

The second method of improving photoactivity is to introduce different functional groups on the ligands of MOFs (such as –$NH_2$ [22] and –OH [23]) to improve light absorption and electron-hole separation efficiency [24]. For instance, Xie et al. designed UiO-66-$(OH)_2$ for the photocatalytic reduction of Cr(VI), which achieved a 100% reduction of Cr(VI) using the high catalytic reduction capacity of nanoscale Zr–O clusters and the visible light photoexcitation capacity of 2,5-dihydroxyterephthalic acid [25]. Shen et al. also successfully synthesized UiO-66-X (X = H, $NH_2$, $NO_2$, Br) and tested its photocatalytic activity for the reduction of Cr(VI) in water treatment, demonstrating that different ligand substituents have significant effects on the photocatalytic activity of UiO-66 [26]. There is a clear structure–photocatalytic activity relationship between the electronic character of the attached substituents and the reaction rate, whereby the electron-donating substituents lead to superior photocatalytic activity of UiO-66, while the electron-absorbing substituents weaken the photoreactivity of UiO-66.

Porphyrin (TCPP) has more carboxyl groups than terephthalic acid, and it is more likely to be protonated for absorbing negatively charged $Cr_2O_7^{2-}$. With visible light irradiation, electrons can transfer from TCPP to the Zr–O cluster, enhancing photo-induced charge carrier separation necessary to reduce Cr(VI). Feng et al. prepared J-aggregated aggregated MOFs using 5,15-di(3,4,5-trihydroxyphenyl)porphyrin reacted with zirconium ions to facilitate the production of singlet oxygen [27]. However, synergistic photocatalytic water treatment between TCPP and UiO-66-$NH_2$ has not yet been reported.

Therefore, a simple solvothermal synthesis method was employed to fabricate the TCPP-modified UiO-66-$NH_2$. The introduction of TCPP enables electrons to transfer from the ligands to the central zirconium–oxygen cluster. The visible light photocatalytic performance test revealed that UNT was highly effective at reducing Cr(VI). Additionally, the influence of factors such as pH and light source were systematically studied. It was finally proposed that Cr(VI) reduction processes may follow a specific mechanism. This work will contribute insights into the fabrication of prospective UiO-66-$NH_2$ photocatalysts to deal with Cr(VI) pollution in water under the sunlight.

## 2. Results and Discussion

### 2.1. XRD Analysis

The crystalline structure of the samples was analyzed by X-ray diffraction (XRD). For the pure UiO-66-NH$_2$, the diffraction peaks at 7.3°, 8.5°, 17.1°, 25.8° and 30.8° correspond to (111), (200), (400), (600) and (711) crystal planes, respectively (Figure 1) [28]. After TCPP was introduced, the characteristic peaks from UNT-5 to UNT-20 were similar to UiO-66-NH$_2$, indicating that TCPP does not affect the structure of the MOF material.

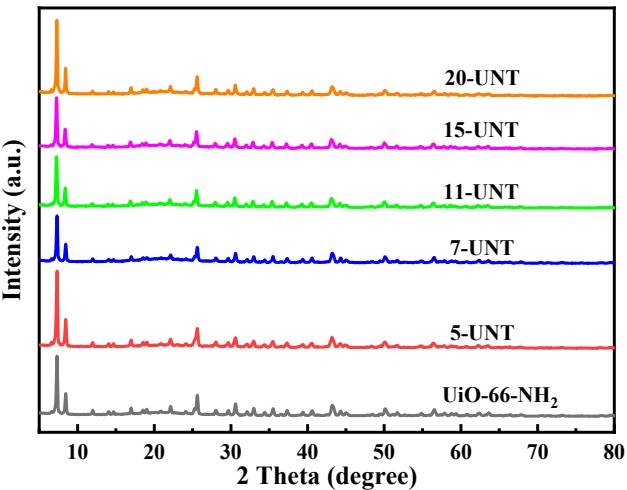

**Figure 1.** XRD patterns of UiO-66-NH$_2$ and UNTs.

### 2.2. FT-IR Analysis

The FT-IR spectra of the prepared photocatalysts were compared in the wave number range of 2000−1000 cm$^{-1}$ (Figure 2). The peak around 1722 cm$^{-1}$ was the result of carboxyl group stretching vibrations in TCPP, and N–H bond stretching vibrations caused the peak at 1606 cm$^{-1}$ [29]. Moreover, the weak band at 1508 cm$^{-1}$ represented the C=C tensile vibration peak in the benzene ring, and the peak around 1657 cm$^{-1}$ was attributed to the –NH$_2$ bending vibrational peak [30]. Additionally, the bands at 1400 cm$^{-1}$ and 1262 cm$^{-1}$ could be assigned to the C–N stretching vibration and C–H bending, respectively. Compared to pure TCPP, the peak at 1722 cm$^{-1}$ disappeared in UNTs, indicating the coordination between –COOH and metal ions. The above results suggest that the TCPP may be successfully loaded on the UiO-66-NH$_2$.

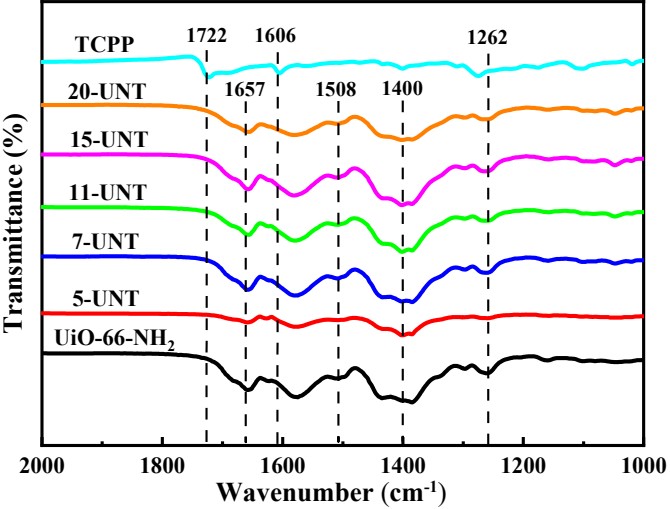

**Figure 2.** FT−IR of the UiO-66-NH$_2$, TCPP and UNTs.

### 2.3. XPS Analysis

The surface elemental compositions and electronic structures of UiO-66-NH$_2$ and 15-UNT were examined by XPS. As shown in Figure 3a, the characteristic peaks at 531.1, 399.1, 284.1 and 182.1 eV correspond to O 1s, N 1s, C 1s and Zr 3d, respectively, which indicated the presence of O, N, C and Zr elements according to the survey XPS spectrum. As shown in Figure 3b, the N 1s spectra of UiO-66-NH$_2$ at 398.8 eV and 15-UNT at 398.9 eV were attributed to the -NH$_2$ group, which indicate the presence of –NH$_2$ in 15-UNT. In addition, as shown in Figure 3c, the characteristic peaks at 184.6 and 182.3 eV correspond to Zr 3d$_{3/2}$ and Zr 3d$_{5/2}$, respectively, and the binding energy of Zr 3d$_{3/2}$ and Zr 3d$_{5/2}$ shifted towards higher values, implying that a Zr–O coordination bond may be formed between UiO-66-NH$_2$ and TCPP. The O 1s spectrum is also shown in Figure 3d. The 15-UNT peak at 530.9 eV was attributed to the O–C bond. After the combination of UiO-66-NH$_2$ with TCPP, the binding energy of the O 1s of the O–C counterpart decreased from 531.1 eV to 530.9 eV, which indicates that the distribution of its electron cloud was changed. All in all, the above results prove the successful preparation of UNT.

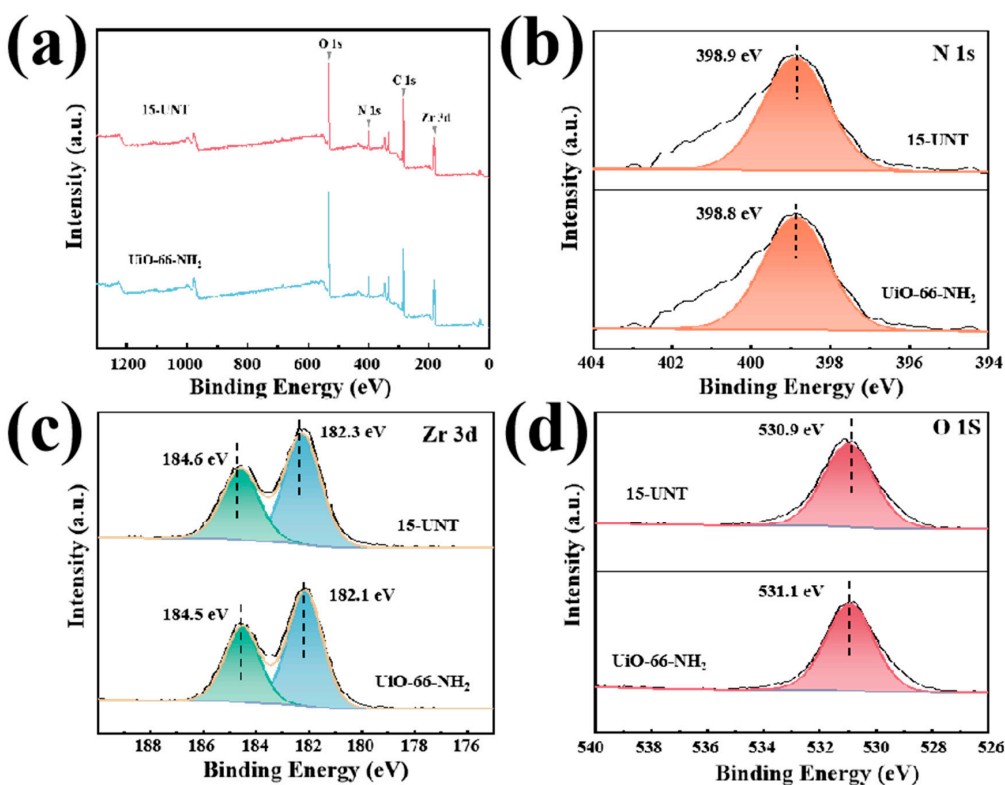

**Figure 3.** (**a**) XPS survey spectrum of UiO-66-NH$_2$ and 15-UNT, (**b**) N 1s spectrum, (**c**) Zr 3d spectrum and (**d**) O 1s spectrum.

### 2.4. UV–Vis Analysis

As shown in Figure 4a, the absorption edge of UiO-66 was at about 300 nm. While terephthalic acid was replaced by NH$_2$-BDC, there was a redshift in the absorption edge of UiO-66-NH$_2$ at 450 nm. The absorption edge of UNT was also significantly shifted to 700 nm in contrast to UiO-66-NH$_2$. Collectively, these results indicate that the light response of the samples could be improved by changing the ligand. The UNT improved the light absorption for longer wavelengths, which correlated with the introduction of TCPP. The UNT retained the specific peaks (Q band) that were assigned to TCPP [31]; this phenomenon indicates that the inner ring of TCPP had not changed. We also found that the light absorption capacity of UNT increased as porphyrin content increased, and, specifically, 20-UNT had the strongest light absorption capacity. The above results prove that TCPP was successfully decorated in UiO-66-NH$_2$. This observation is also consistent with the

XRD and FT-IR of the samples. Moreover, the band gap energies (Eg) were computed with the Tauc plots (Figure 4b). Here, the UiO-66-NH$_2$ and 15-UNT were estimated to be about 2.92 eV and 2.68 eV, respectively. These values are in agreement with the values reported in other works [32].

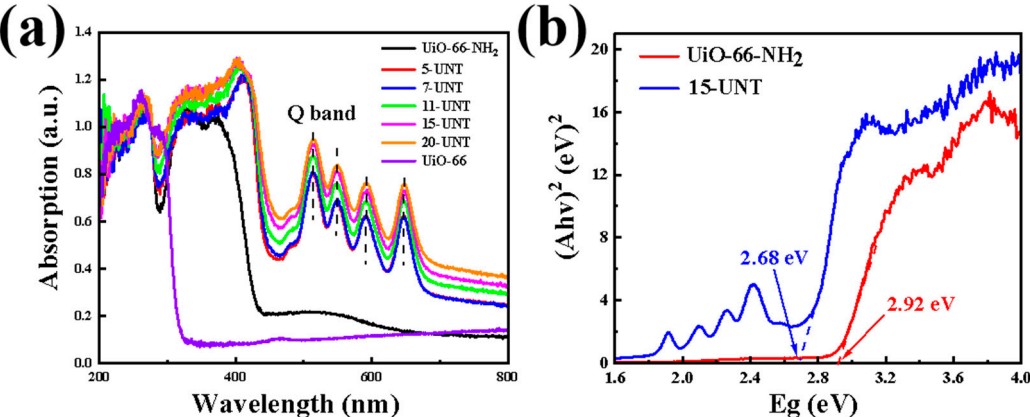

**Figure 4.** (**a**) UV–Vis spectra of the samples and (**b**) the bandgap energy.

### 2.5. SEM Analysis

According to the scanning electron microscope images, UiO-66-NH$_2$ appeared with octahedral morphology and particle sizes of about 100 nm (Figure 5), which was similar to a previous report [33]. As shown in Figure 5a–c, the particle size of the crystals remains constant at around 100 nm when TCPP is added in amounts below 11 mg, and the apparent octahedral "cone" form can be seen.

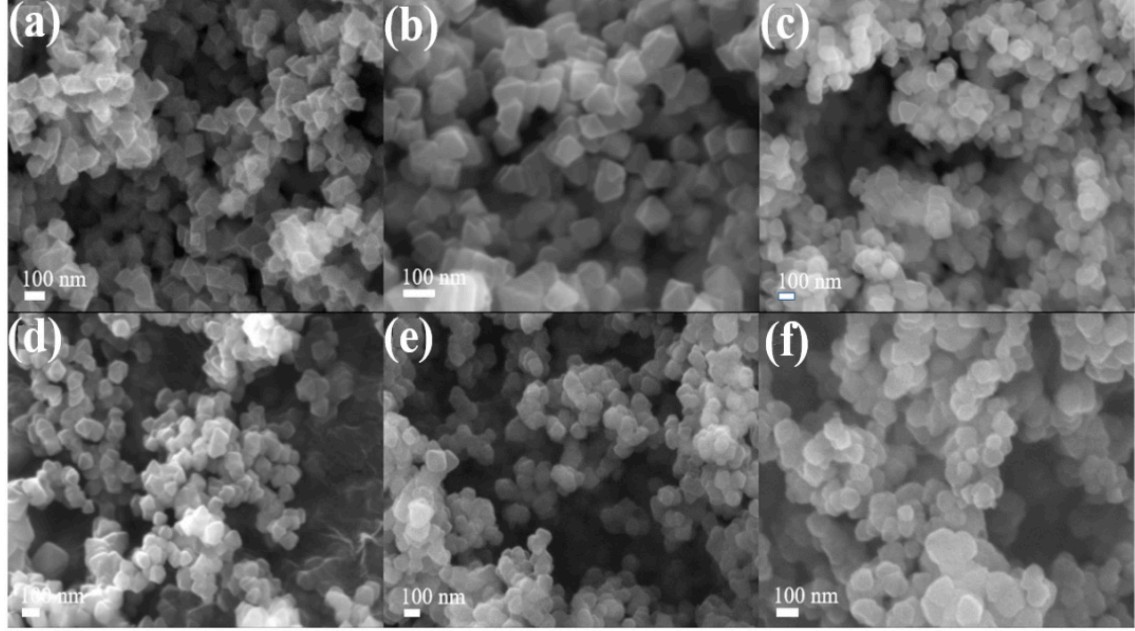

**Figure 5.** SEM of (**a**) UiO-66-NH$_2$, (**b**) 5-UNT, (**c**) 7-UNT, (**d**) 11-UNT, (**e**) 15-UNT and (**f**) 20-UNT.

As shown in Figure 6a–c, the particle size distribution plots showed that the particle sizes of UiO-66-NH$_2$, 5-UNT and 7-UNT were roughly 100 nm, 103 nm and 104 nm, respectively, with minor variations in particle size. When the content of TCPP was greater than 11 mg, the morphology of UNT began to slowly deform, the "rounded" crystal structure started to appear, and the grain size tended to become larger (Figure 5d–f). In addition, as shown in Figure 6d–f, the particle size distribution plots can be seen to be roughly 108 nm, 110 nm and 123 nm for 11-UNT, 15-UNT and 20-UNT, respectively. Lastly, the 20-UNT displayed an irregular shape and its particle sizes increased.

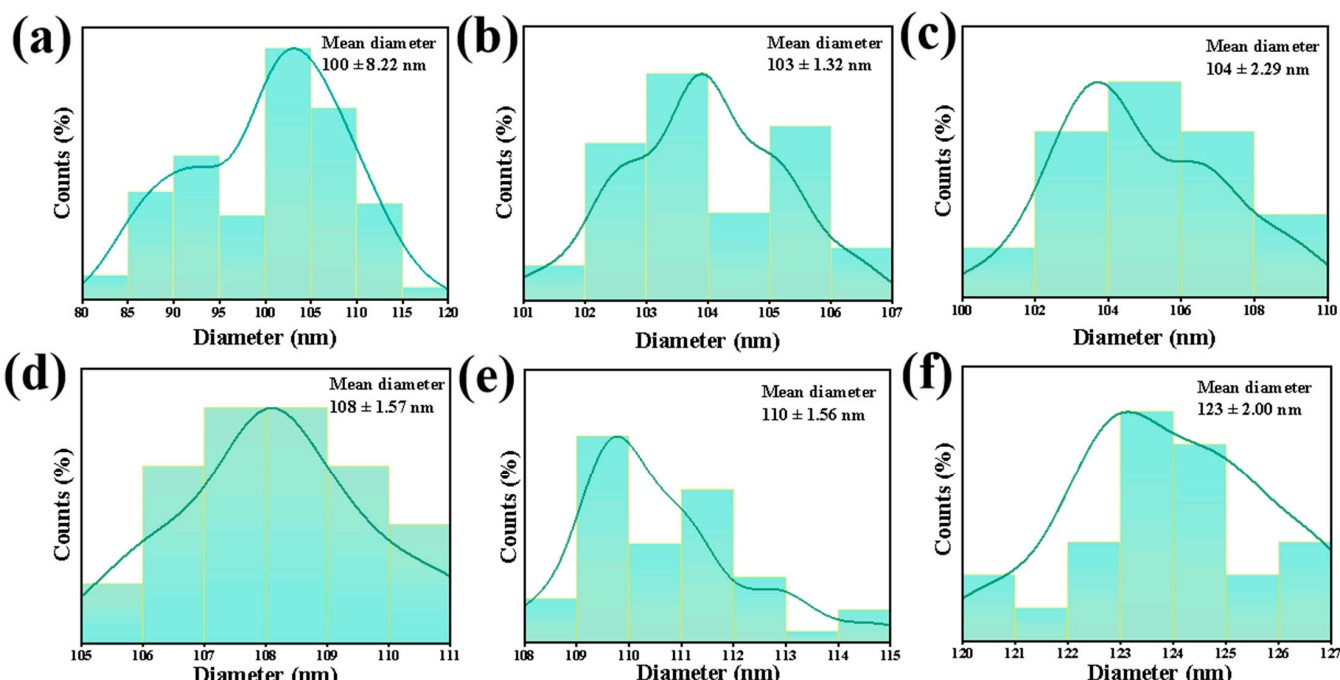

**Figure 6.** Particle size distribution of (**a**) UiO-66-NH$_2$, (**b**) 5-UNT, (**c**) 7-UNT, (**d**) 11-UNT, (**e**) 15-UNT and (**f**) 20-UNT.

### 2.6. Nitrogen Sorption Analysis

A type I curve was found to exist for both 15-UNT and UiO-66-NH$_2$ samples at 77 K (Figure 7a), implying that the samples possessed a microporous structure. With the addition of TCPP to UiO-66-NH$_2$, the adsorption isotherm still maintained the type I isotherm, but the BET-specific surface area of 15-UNT decreased from 852.6 to 736.8 m$^2$/g as compared to pure UiO-66-NH$_2$. A possible reason to explain this observation is that the loading of TCPP increased the surface roughness and reduced the smoothness of the surface, which decreased the specific surface area. However, the specific surface area of 15-UNT did not change significantly, indicating that the introduction of TCPP had little effect on the specific surface area. Furthermore, the existence of a microporous structure in the 15-UNT was confirmed by the pore size distribution plots (Figure 7b).

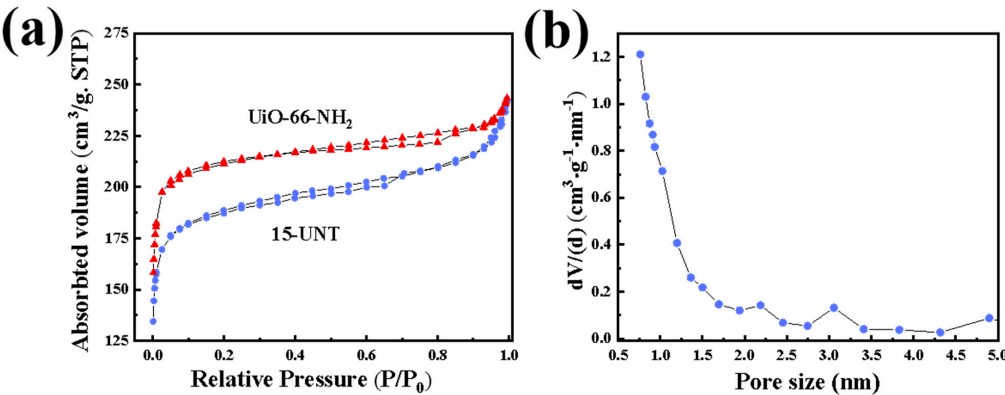

**Figure 7.** (**a**) N$_2$ adsorption–desorption isotherm of UiO-66-NH$_2$ and 15-UNT. (**b**) Pore size distribution plots of 15-UNT.

### 2.7. Photocatalytic Performance

UNT was investigated for its photocatalytic performance by reducing 100 ppm Cr(VI) with full light illumination (Figure 8). Theoretically, the adsorption equilibrium should be reached before the photocatalytic reaction; therefore, we performed 180 min dark ad-

sorption experiments on the 15-UNT and found that it reached the adsorption equilibrium point at 60 min (Figure 8a). From the 5-UNT to the 15-UNT, with increasing TCPP content, Cr(VI) photocatalytic reduction performance improved (Figure 8b). However, the 20-UNT did not increase significantly compared with the 15-UNT. Therefore, in this experiment, we further investigated the sample performance of 15-UNT.

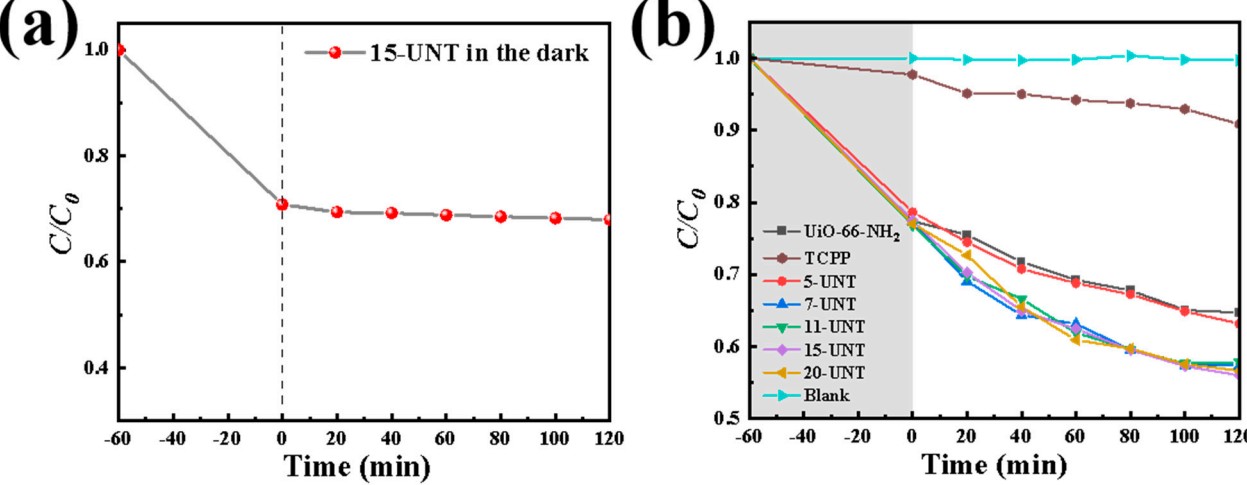

**Figure 8.** (**a**) 15−UNT dark adsorption of Cr(VI). (**b**) Photocatalytic reduction performance for 100 ppm Cr(VI) under full light.

　　Notably, it had been reported in the literature that the reduction of Cr(VI) was susceptible to the chemical environment of the system, especially to changes in pH [34]. Therefore, we used 15-UNT to reduce Cr(VI) in a photocatalytic manner under different pH values (tested pH = 1, 3, 5, 7 and system's original pH). With the increase of acidity, Cr(VI) became more efficient for photocatalytic reduction (Figure 9a). It was observed that Cr(VI) was nearly reduced at the 80 min time point at pH = 1. Additionally, under the pH = 1 condition, the 15-UNT had the largest slope and the kinetic constant, k, was $5.9 \times 10^{-4}$ min$^{-1}$, which was 18.43 times the constant value at pH = 3 and 29.5 times the value at original pH (Figure 9b,c). The gradient color change of the system during the reaction is shown in Figure 9d. From 0 min to 80 min, the color gradually became lighter. In the light condition at 80 min, the solution became colorless, indicating that the 15-UNT has an excellent reduction property for hexavalent chromium.

　　The light waves below 420 nm are filtered out with a filter and then the property of the catalyst is measured. Under visible light irradiation, 15-UNT still maintained excellent photocatalytic reduction performance, reaching a degradation rate of 100% within 120 min (Figure 10a,c). The value of the kinetic constant, k, for 15-UNT was 10 times that of UiO-66-NH$_2$ (Figure 10b,d). Additionally, the experimental study showed that the photoreduction capacity could be improved by introducing TCPP in the UiO-66-NH$_2$ system.

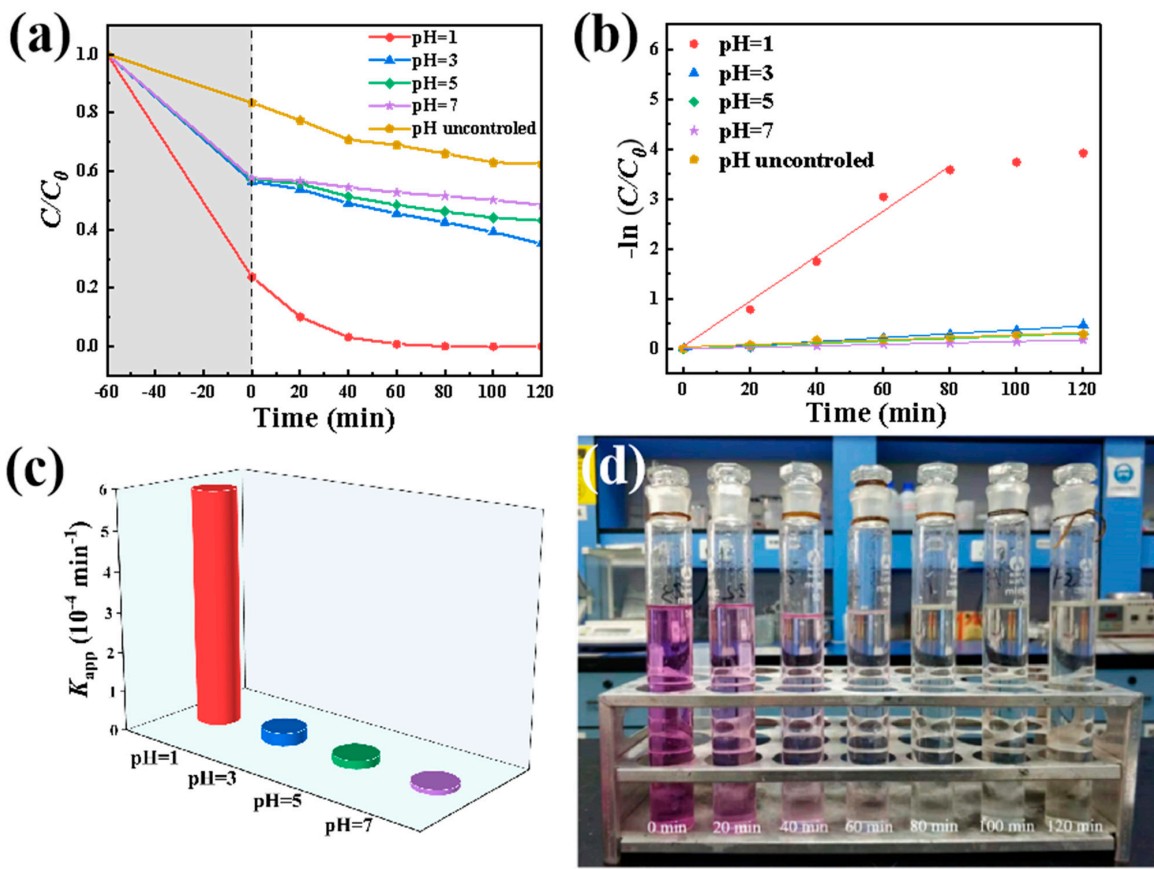

**Figure 9.** (**a**) 15−UNT photocatalytic reduction of Cr(VI) at different pH values and (**b**,**c**) the corresponding kinetics; (**d**) reaction process diagram by photo.

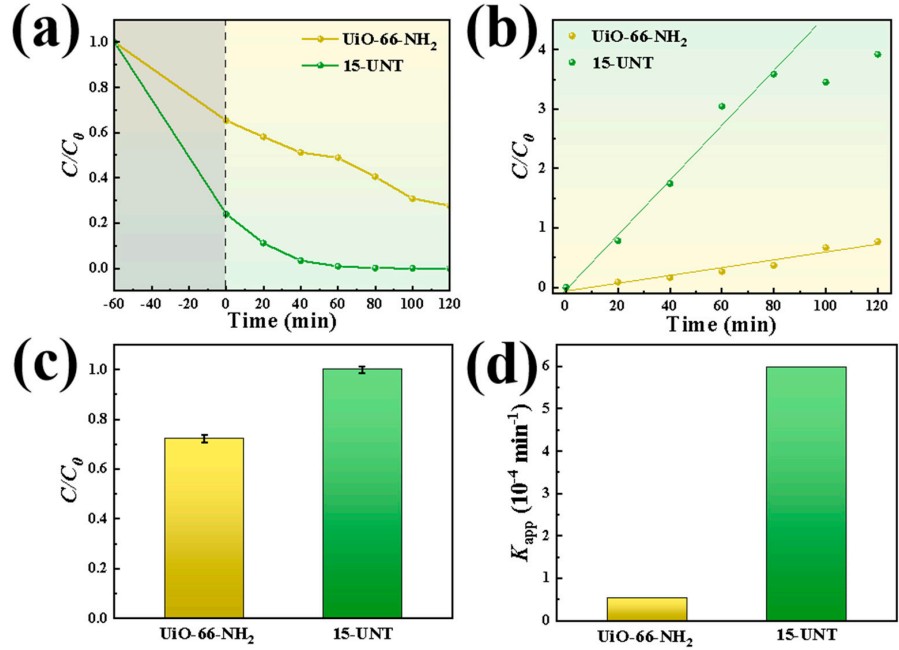

**Figure 10.** (**a**,**c**) Performance of photoreduction of hexavalent chromium at wavelengths above 420 nm and (**b**,**d**) the corresponding kinetics.

In order to investigate the removal of Cr(VI) by 15-UNT under visible light, XPS analysis of 15-UNT was carried out before and after the reaction. As shown in Figure 11a, the survey XPS spectrum of the reacted samples appeared as characteristic peaks at 576.1, 531.1, 399.1, 284.2 and 182.2 eV, which corresponded to Cr 2p, O 1s, N 1s, C 1s and Zr 3d, respectively. This was followed by deconvolution integration peaks for Cr 2p, which corresponded to Cr $2p_{3/2}$ at 576.9 and 580.2 eV and to Cr $2p_{1/2}$ at 586.7 and 589.6 eV (Figure 11b). In addition to these findings, the characteristic peaks at 576.9 and 586.7 eV respond to Cr(III), while the peaks at 580.2 and 589.6 eV correspond to Cr(VI). The above results demonstrate the simultaneous presence of Cr(VI) and Cr(III) on the surface of 15-UNT after photocatalytic reduction as well as the presence of the reduction of Cr(VI) to Cr(III).

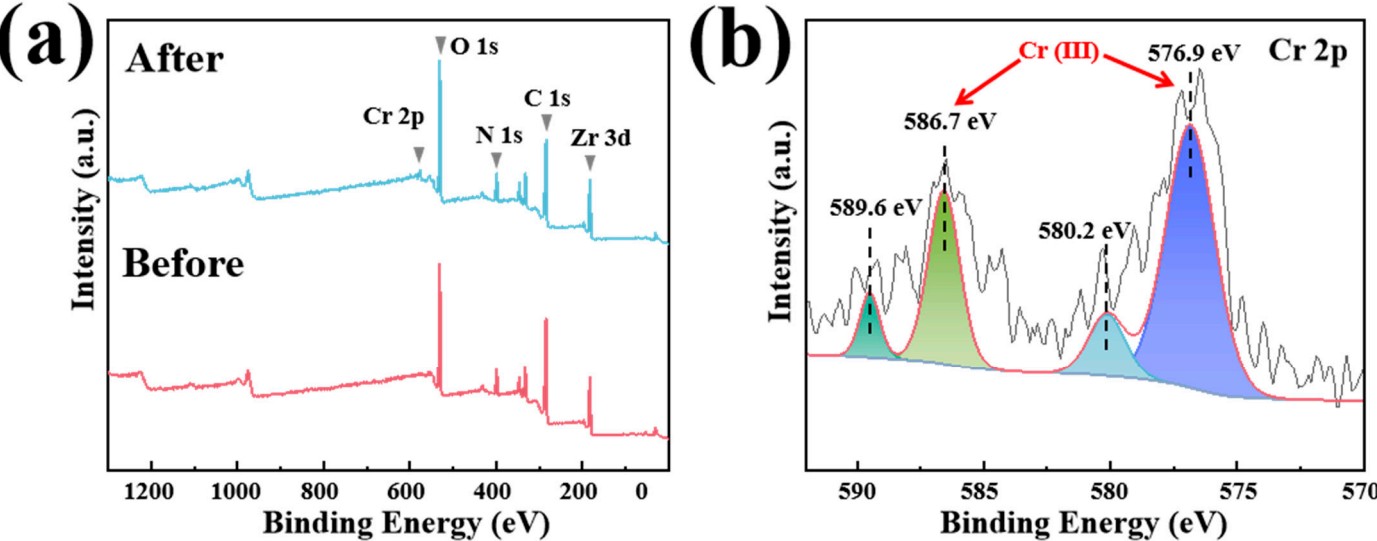

**Figure 11.** (**a**) XPS survey spectrum before and after Cr(VI) removal experiments of 15-UNT. (**b**) Cr 2p spectrum of 15-UNT after experiments.

Moving on, Figure 12 shows the reusability and stability of the 15-UNT. There was no significant decrease in the second and third rounds of cycle experiments, indicating that 15-UNT has a stable photocatalytic reduction performance (Figure 12a). Simultaneously, the stability of 15-UNT was further confirmed by XRD, SEM, FT-IR and UV–Vis analyses (Figure 12b–f). As shown in Figure 12b, the XRD pattern of 15-UNT before and after the reaction showed that the crystal structure did not change appreciably. The diffraction peaks at 7.3°, 8.5°, 17.1°, 25.8° and 30.8° correspond to the (111), (200), (400), (600) and (711) crystal planes of 15-UNT, respectively. As can be seen in Figure 12c,d, after the three cycles, the morphology of 15-UNT remained stable. Furthermore, as shown in Figure 12e, it was found that the -NH$_2$, N–H, C=C, C–N and C–H absorption peaks located at 1657 cm$^{-1}$, 1606 cm$^{-1}$, 1508 cm$^{-1}$, 1400 cm$^{-1}$ and 1262 cm$^{-1}$, respectively, did not change significantly through the FT-IR spectra of 15-UNT before and after the reaction. The 15-UNT also retained the specific peaks (Q band) that were assigned to TCPP. Figure 12f showed the leaching of TCPP before and after the reaction, with no changes in the position of the characteristic absorption peak in the Q band of TCPP and without any significant intensity decrease. Collectively, the results here show that the structure of 15-UNT did not change before and after the reaction.

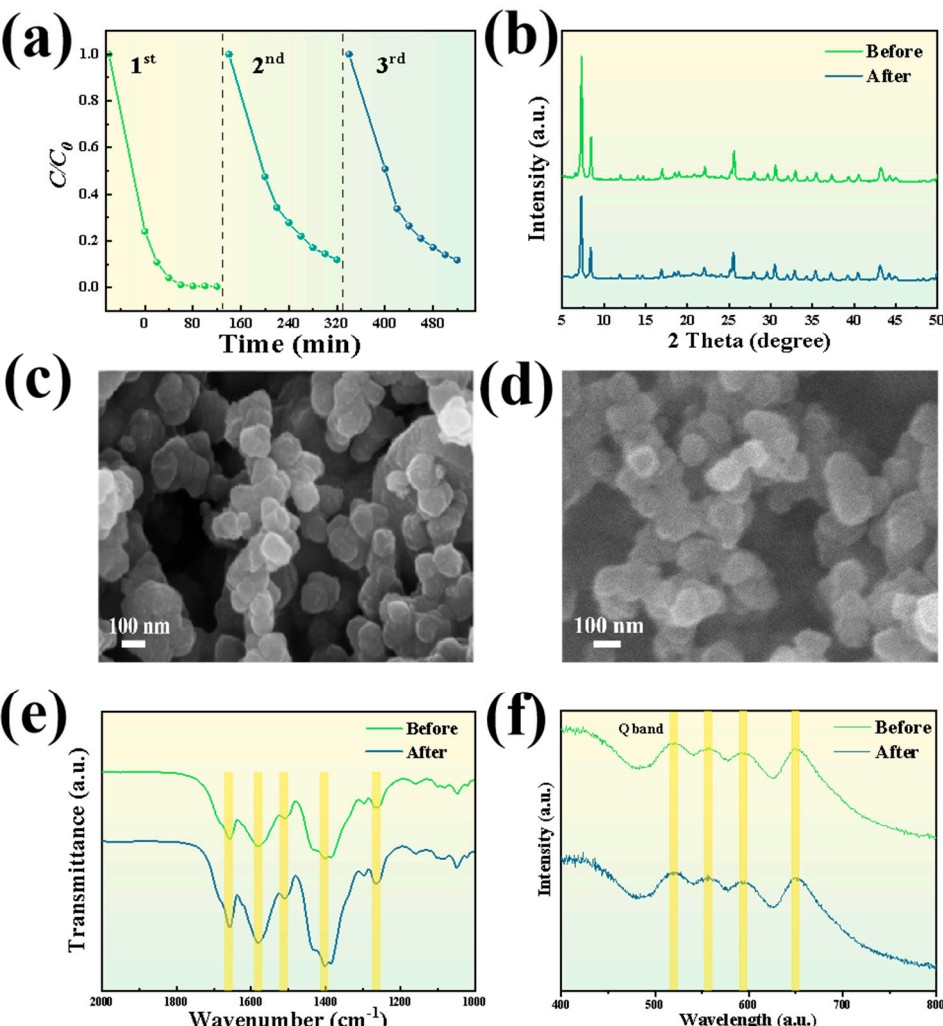

**Figure 12.** (**a**) Cycling experiments of 15−UNT under visible light irradiation. (**c,d**) SEM image of 15−UNT after reaction; (**b**) XRD, (**e**) FT−IR and (**f**) UV−Vis plots of 15−UNT before and after the reaction.

## 3. Photocatalytic Reduction Mechanism

Photoluminescence (PL) measurements were useful for understanding the recombination of photogenerated carriers. The higher the photoluminescence intensity, the higher the efficiency of the carrier complex efficiency. According to Figure 13a, the maximum emission for UiO-66-NH$_2$ was observed at around 450 nm. Comparatively, the PL intensity of 15-UNT showed a significant decrease, which indicates that the carrier recombination rate is lower than UiO-66-NH$_2$. Time-resolved photoluminescence (TRPL) spectroscopy was applied to monitor charge carrier dynamics (Figure 13b). The fluorescence lifetimes of UiO-66-NH$_2$ and 15-UNT were obtained by second-order fitting, as shown in Table 1. UiO-66-NH$_2$ has two time constants, which are $\tau_1$ = 1.91 ns and $\tau_2$ = 9.63 ns. The two time constants for 15-UNT are $\tau_1$ = 1.36 ns and $\tau_2$ = 9.69 ns, respectively. The $\tau_{avg}$ (average lifetimes of the carriers) for UiO-66-NH$_2$ and 15-UNT were calculated to be 1.44 and 3.98 ns, respectively. The results confirmed that the decay time of 15-UNT was faster than that of UiO-66-NH$_2$ and that the photogenerated carriers of 15-UNT had a longer lifetime, suggesting that the loading of TCPP was beneficial in enhancing the photocatalytic activity of UiO-66-NH$_2$.

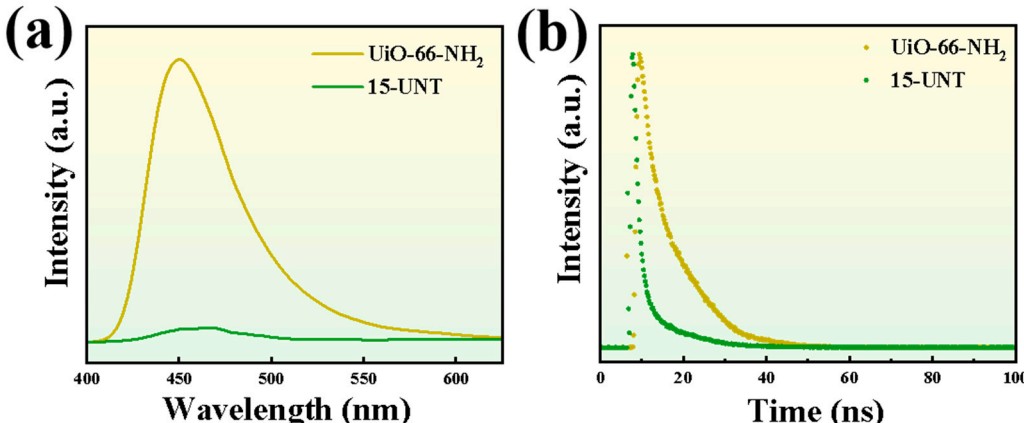

**Figure 13.** (**a**) Photoluminescence spectra. (**b**) Time-resolved photoluminescence spectra.

**Table 1.** Fluorescence lifetime for UiO-66-NH$_2$ and 15-UNT.

| Sample | $\tau_1$ | $\tau_2$ |
|---|---|---|
| UiO-66-NH$_2$ | 1.91 ns | 9.63 ns |
| 15-UNT | 1.36 ns | 9.69 ns |

The photocurrent test showed that 15-UNT had a stronger photocurrent intensity than UiO-66-NH$_2$ (Figure 14a), which indicates that the 15-UNT has a better carrier separation efficiency under light illumination. Usually, a smaller arc radius represents a smaller impedance. In this study, the 15-UNT possessed a smaller arc radius than UiO-66-NH$_2$ (Figure 14b), suggesting that the 15-UNT has a weaker charge transfer impedance and that separation of electrons from holes is more likely to occur.

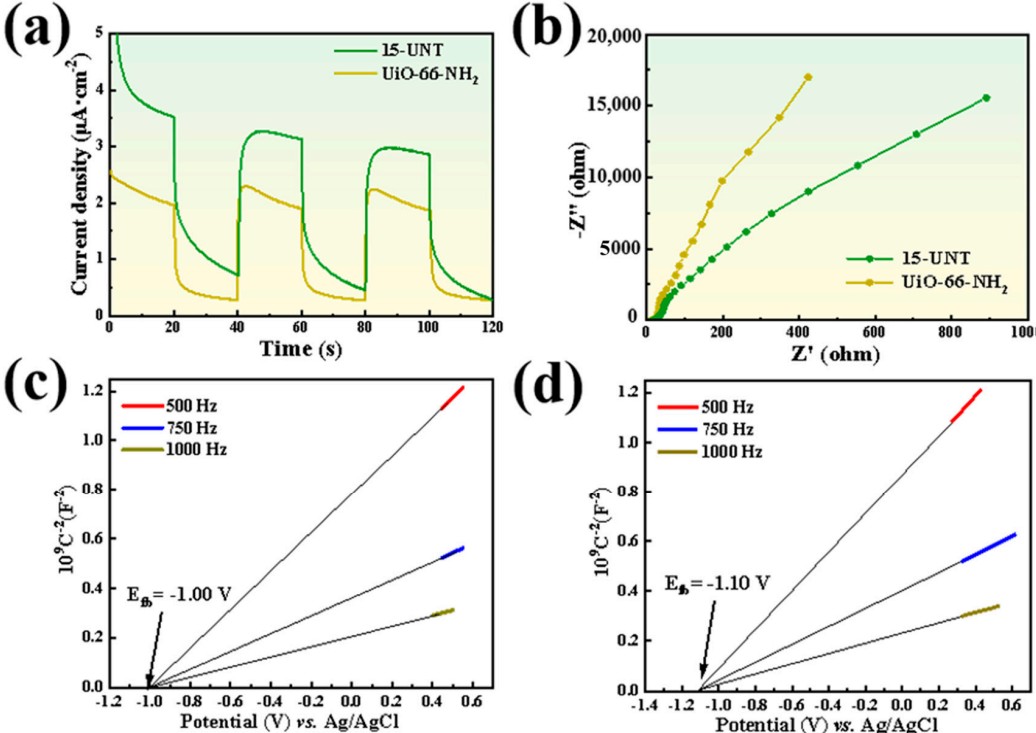

**Figure 14.** Photoelectricity test of UiO$-$66$-$NH$_2$ and 15$-$UNT: (**a**) transient photocurrent density, (**b**) EIS spectra, (**c**,**d**) various frequency MS.

Furthermore, through Mott–Schottky (MS) tests at different frequencies, the flat band potential was calculated for the samples (Figure 14c,d). C$^{-2}$ and potential were

found to be positively correlated, meaning that the photocatalysts are n-type semiconductors. In addition, the flat band potentials were determined for UiO-66-NH$_2$ and 15-UNT to be $-1.00$ V and $-1.10$ V, respectively. In accordance with the formula E(NHE) = E(Ag/AgCl) + 0.0591 × pH + 0.1976 [35], E(Ag/AgCl) was the flat-band potential measured in this experiment. In this case, UiO-66-NH$_2$ and 15-UNT at the normal hydrogen electrode were found at $-0.40$ V and at $-0.50$ V, respectively. Due to the fact that the flat band potential for n-type semiconductors is positive 0.2 V over the conduction band potential [35], the CB of UiO-66-NH$_2$ and 15-UNT were calculated for $-0.60$ V and $-0.70$ V, respectively. These conduction bands were much higher than the reduction potential of Cr(VI). Based on the formula VB = CB + E$_g$ [35], VB values for UiO-66-NH$_2$ and 15-UNT were calculated to be 2.32 V and 1.98 V, respectively.

On the basis of the above analysis, the possible explanation for hexavalent chromium reduction by the UNT via photocatalysis was provided in Figure 15. Under acidic conditions, the amino group of UiO-66-NH$_2$ and the carboxy group of TCPP will be protonated, which means the UNT will be positively charged. Furthermore, the UNT will adsorb the negatively charged Cr(VI) through electrostatic attraction. Meanwhile, the TCPP and UiO-66-NH$_2$ can produce photogenerated electrons to reduce Cr(VI) under light irradiation. For the TCPP, its photogenerated electrons have a higher potential than UiO-66-NH$_2$. This result indicates that a portion of the electrons can transfer to UiO-66-NH$_2$ through the ligand. At the same time, the above electron transfer process promotes the electron-hole separation on the UNT, thus facilitating photocatalytic Cr(VI) reduction.

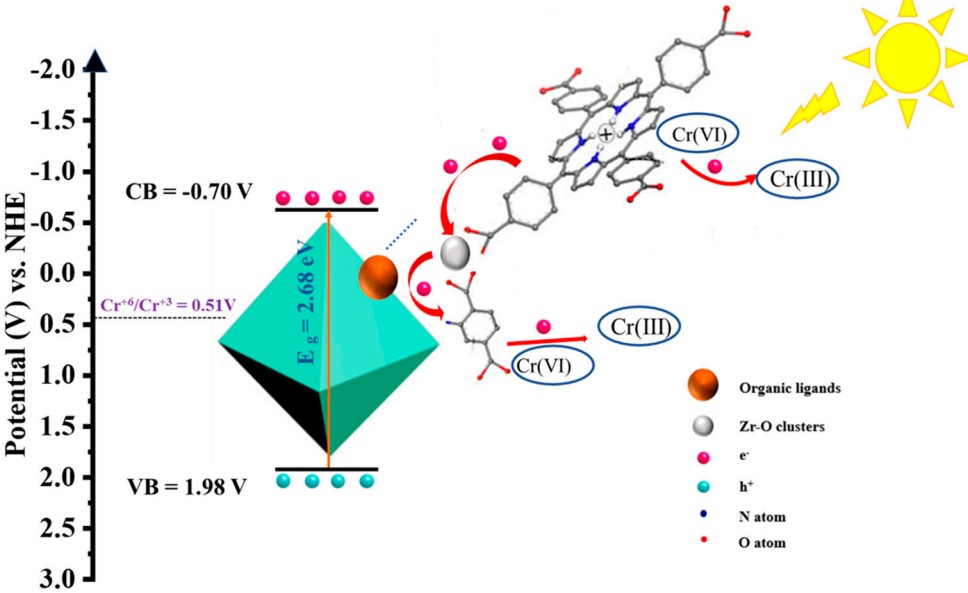

**Figure 15.** Schematic representation of the mechanism of Cr(VI) reduction by UNT under visible light.

## 4. Materials and Methods

### 4.1. Chemicals and Reagents

NH$_2$-BDC (C$_8$H$_7$NO$_4$, 98.0%) was purchased from Aladdin Reagent Co., Ltd. (Shanghai, China). Zirconium chloride (ZrCl$_4$, 98.0%), terephthalic acid (C$_8$H$_6$O$_4$, 99.0%), DMF (C$_3$H$_7$NO, 99.0%), acetate (CH$_3$COOH, 99.0%), methanol (CH$_3$OH, 99.0%), porphyrin (TCPP, C$_{48}$H$_{30}$N$_4$O$_8$, 97.0%) and diphenylcarbazide (C$_{13}$H$_{14}$N$_4$O) were purchased from Macklin Chemistry Co., Ltd. (Shanghai, China), while Phosphoric acid (H$_3$PO$_4$, AR), Sulfuric acid (H$_2$SO$_4$, AR) and Sodium hydroxide (NaOH, AR) were obtained from the Beijing Chemical Industry Group Co., Ltd. (Beijing, China). All these chemicals were directly used without further purification.

### 4.2. Synthesis of UiO-66-NH$_2$ and x-UNT

UiO-66-NH$_2$ was synthesized based on the existing literature [36]. In DMF (6.0 mL), ZrCl$_4$ (50.0 mg) and organic ligand NH$_2$-BDC (51.0 mg) were dissolved. Subsequently, in an ultrasonic bath, the solution was added with 1.1 mL of acetic acid for 30 min, followed by further ultrasonic treatment. After this was completed, an autoclave lined with Teflon was used to bottle the solution, followed by an overnight heating process at 120 °C. As soon as the reaction was complete, the solution was cooled to room temperature. After centrifugation for three times with DMF and absolute methanol, the sample was dried at 60 °C for 12 h in a vacuum oven.

The UNT was prepared by using 5 mg, 7 mg, 11 mg, 15 mg and 20 mg TCPP in the DMF solution, respectively. Thereafter, NH$_2$-BDC (51.0 mg) and ZrCl$_4$ (50.0 mg) were added to the solutions, and the subsequent operation was the same as UiO-66-NH$_2$. The composites were denoted as 5-UNT, 7-UNT, 11-UNT, 15-UNT and 20-UNT, respectively.

### 4.3. Photocatalytic Measurements

The K$_2$Cr$_2$O$_7$ was used as a simulated pollutant in the photocatalytic reduction experiments. Typically, Cr(VI) model was made by 25.0 mg photocatalysts mixed with 50.0 mL 100.0 mg L$^{-1}$ K$_2$Cr$_2$O$_7$ aqueous solution. By adding H$_2$SO$_4$ or NaOH solutions, the pH of the solution was adjusted. Moreover, the photocatalytic performance tests were carried out under the illumination of a 500 W xenon lamp (XE-JY500, Beijing NBET Technology Co., Ltd., Beijing, China), and its filter had a wavelength cut-off of 420 nm. The system was exposed to light with stirring after adsorption–desorption equilibrium had been achieved. Following this, 5.0 mL of solution was extracted every 20 min. A supernatant of 2 mL was collected for further analysis after centrifuging at 8000 rpm for 5 min. The method of diphenyl carbazide (DPC) was used to determine the concentration of the residual Cr(VI) [37], and the photocatalytic reaction efficiency was obtained via Equation (1).

$$\eta = C/C_0 \tag{1}$$

where η is photocatalytic performance efficiency, C$_0$ is the initial absorbance of the pollutant and C is the absorbance of pollutants at different times.

A pseudo-first-order kinetic model was calculated via Equation (2) to make further comparisons with the photocatalytic efficiencies.

$$\ln(C_0/C) = kt \tag{2}$$

where C$_0$ and C are the initial concentration and the remaining concentration of RhB or TC at each time point, respectively. k is the kinetics rate constant, and t is the reaction time.

### 4.4. Characterization

The crystalline properties of all as-synthesized samples were determined via X-ray diffraction (XRD, Bruker D8 Advance, Billerica, MA, USA) with a Cu Kα radiation source (λ = 0.15406 nm) in the 2θ range of 5–80° at a rate of 2°·min$^{-1}$. An FT-IR spectrometer (PerkinElmer Spectrum 100, Waltham, MA, USA) was used to obtain information about the function groups. In addition, scanning electron microscopy (SEM, Field Emission Scanning Electron Microscope SU8010, Tokyo, Japan) was used to examine the morphologies of the samples. Moreover, the optical properties and band gap energies were determined via UV–VIS diffuse reflectance spectra (DRS, Shimadzu UV-3900, Kyoto, Japan), and the photoluminescence (PL) spectra of all photocatalysts were investigated by using a steady-state/transient fluorescence spectrometer (F-4700, Hitachi, Tokyo, Japan). The excitation wavelength was 365 nm. Lastly, in an Autosorb-1 nitrogen adsorption apparatus from Quantachrome, the surface areas were analyzed with the Brunauer–Emmett–Teller (BET) method.

### 4.5. Electrochemical Tests

The electrochemical tests were measured on a widespread three electrode mode by the electrochemical workstation (Huachen CHI-760E, Shanghai, China), and $Na_2SO_4$ solution with a concentration of 0.1 M was used as the electrolyte. The photocurrent response was recorded by using a 500 W xenon lamp (Solar-500, Beijing NBET Technology Co., Ltd., Beijing, China) equipped with a 420 nm cutoff filter. The working electrode was prepared as follows: 10 mg of catalysts was first mixed with 1 mL of DI water to produce a slurry. For counter electrodes, platinum electrodes were used; whereas, for reference electrodes, saturated Ag/AgCl electrodes were used. A range of voltages between $-0.6$ and 0.8 V was measured with Mott–Schottky tests. EIS tests were also conducted at the frequencies of 500, 750 and 1000 Hz.

### 5. Conclusions

In summary, porphyrin-modified UiO-66-NH$_2$ was obtained via a solvothermal process, which enhanced the light adsorption ability and the photocatalytic efficiency. In addition, the photocatalytic performance remains essentially unchanged under visible and full light irradiation. We have also investigated the removal of Cr(VI) at different pH values. Under the visible light irradiation and pH = 1, the reduction rate of hexavalent chromium by 15-UNT is 10 times greater than that of the original UiO-66-NH$_2$ and eliminates 100% of the Cr(VI) within 80 min. Based on the test of PL and the electrochemical analysis, the introduction of porphyrin in UiO-66-NH$_2$ facilitates the photo charge carriers' migration and separation. Overall, this work provides a reference for the modification of MOF photocatalysts from the perspective of ligands and provides additional insights into the study of the synergistic photocatalytic performance of adsorption and photosensitization.

**Author Contributions:** Conceptualization, K.Y., D.C., B.G. and C.P.; methodology, B.G. and C.P.; software, K.Y., B.G. and C.P.; validation, K.Y., D.C. and B.G.; formal analysis, B.G. and C.P.; investigation, K.Y.; resources, D.C., Y.H. and D.H.; data curation, K.Y., B.G. and C.P.; writing—original draft preparation, K.Y. and B.G.; writing—review and editing, K.Y., B.G., Y.F. and K.C.; supervision, D.C. and Y.H.; project administration, D.C. and Y.H.; funding acquisition, D.C. All authors have read and agreed to the published version of the manuscript.

**Funding:** This work was supported by the National Natural Science Foundation of China (No. 21978276) and the Fundamental Research Funds for the Central Universities (No. 2652019157, 2652019158, 2652019159).

**Data Availability Statement:** Not applicable.

**Conflicts of Interest:** The authors declare no conflict of interest.

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
