# Peer review of "Porphyrin Modified UiO-66-NH2 for Highly Efficient Photoreduction of Cr(VI) under Visible Light"

_catalysts, doi:10.3390/catal13071073_

Round 1
Reviewer 1 Report
The manuscript titled “Porphyrin modified UiO-66-NH2 for highly efficient photoreduction of Cr(IV) under visible light” presented by Kaiwen Yuan, Bo Gong, Chundong Peng, Yanmei Feng, Yingmo Hu, Kai Chen, Daimei Chen, and Derek Hao deals with the photocatalytic measurements and characterization of catalysts for conversion of Cr(VI) to Cr(III). However, there are issues that could be clarified:
- Abbreviation TCPP appeared in the Abstract should be explained.
- Line 101: “UNT-5 to UNT-20” are not explained.
- UV-vis section: The bang gap energy is calculated using Tauc equation, not Kubelka-Munk method. The BG energy for 15-UNT is incorrect, it should be about 2.7 eV (according to Tauc plot).
- Figure 7a has incorrect label.
- Section 2.6: The author demonstrated the influence of pH on the photocatalytic process. Could you clarify the pH of solution presented on Figure 7b?
There are major issue concerning the manuscript:
1. The author presented the photocatalysts for removal of Cr(IV) cations from the wastewater (pH = 5-6). But the best activity of photocatalyst is reached at pH=1 when at pH=5-6 the activity is slightly higher that the activity of UiO-66-NH2. Could the author explain the future use of the photocatalysts in the conditions close to real ones?
2. Could the author discuss the final conversion state of Cr(VI) cations? Cr2O3? What is the form of converted Cr(VI) – residue or adsorbed species on the UiO surface?
3. Figure 13 (Scheme) is incorrect taking into account the incorrect measurement of band gap for UTN-15.
4. To support the conclusion (conversion of Cr(VI) to Cr(III)) it is necessary to do the XPS measurement the Cr2p after reaction.
The presentation of study should be revised.
It could be improved.
Reviewer 2 Report
The paper is devoted to the study of MOF/porphyrine composite in the photodegradation of highly toxic and carcinogenic Cr(VI) species. The overall research idea is not very novel. The introduction is well written and presents the topic and previous achievements in this field sufficiently. The basic characterization of the samples does not raise any question. However, some important corrections are needed to prove the presented catalytic efficiencies.
1. All the inital data (i.e. UV/vis spectra) must be provided to support figures 7-9. The only pure (C/C0) values derived from these spectra can not be evaluated correctly.
2. A test on porphyrine leaching from the catalyst needs to be performed.
3. Simiar Cr(VI) photoreduction tests should be performed for TCPP solutions without MOF. What is a point of using MOF as a component in this system?
4. Finally, a form in which TCPP exists within the MOF should be analyzed. Is it adsorbed on the surface of crystallites, incorporated as a counteranion into the pores, coordinated to metal center, or bound to UiO NH2 groups through amide bonds? Is it distributed unformly? Can Zr(4+) coordinate into the porphyrinic ring during the synthesis? Such information is very important to prove the reproducibilty of results and to support the mechanism speculations.
Minor english style and grammar revisions are necessary.
Round 2
Reviewer 1 Report
The authors have carried out an additional study, corrected the manuscript according to comments.
Minor editing of English language required
Reviewer 2 Report
Authors have carefully addressed all the reviewers' comments. I recommend the acceptance of the paper.